# Consciousness Detection in a Complete Locked-in Syndrome Patient through Multiscale Approach Analysis

**DOI:** 10.3390/e22121411

**Published:** 2020-12-15

**Authors:** Shang-Ju Wu, Nicoletta Nicolaou, Martin Bogdan

**Affiliations:** 1Neuromorphic Information Processing, Leipzig University, Augustusplatz 10, 04109 Leipzig, Germany; bogdan@informatik.uni-leipzig.de; 2Department of Basic and Clinical Sciences, University of Nicosia Medical School, 93 Agiou Nikolaou Street, Engomi 2408, Nicosia, Cyprus; nicolaou.nic@unic.ac.cy; 3Centre for Neuroscience and Integrative Brain Research (CENIBRE), University of Nicosia Medical School, 93 Agiou Nikolaou Street, Engomi 2408, Nicosia, Cyprus

**Keywords:** consciousness, electrocorticography (ECoG), completely locked-in state (CLIS), multiscale approach, sample entropy, permutation entropy, Poincaré plot

## Abstract

Completely locked-in state (CLIS) patients are unable to speak and have lost all muscle movement. From the external view, the internal brain activity of such patients cannot be easily perceived, but CLIS patients are considered to still be conscious and cognitively active. Detecting the current state of consciousness of CLIS patients is non-trivial, and it is difficult to ascertain whether CLIS patients are conscious or not. Thus, it is important to find alternative ways to re-establish communication with these patients during periods of awareness, and one such alternative is through a brain–computer interface (BCI). In this study, multiscale-based methods (multiscale sample entropy, multiscale permutation entropy and multiscale Poincaré plots) were applied to analyze electrocorticogram signals from a CLIS patient to detect the underlying consciousness level. Results from these different methods converge to a specific period of awareness of the CLIS patient in question, coinciding with the period during which the CLIS patient is recorded to have communicated with an experimenter. The aim of the investigation is to propose a methodology that could be used to create reliable communication with CLIS patients.

## 1. Introduction

The typical characteristic of locked-in syndrome (LIS) patients is nearly complete paralysis while retaining full cognition. Car accidents, strokes or motor neuronal diseases, such as amyotrophic lateral sclerosis (ALS), are some examples of conditions that could lead to a LIS state. Patients in a LIS state are often misdiagnosed as suffering from a disorder of consciousness. One such example is a patient who was regarded as being in an unresponsive wakefulness syndrome (UWS) state until 20 years later, when the patient woke up [1]. LIS patients can communicate with the outside world by moving their eye muscles or eyebrows. However, when the patients slip into the completely locked-in state (CLIS), they eventually lose control of these last few remaining muscles, such as the anal sphincter and eye movements or eyebrows [2], but their cognition is assumed to remain intact.

Despite constantly evolving technology, the ground truth of the level of consciousness of CLIS patients is still missing. Some researchers attempted to communicate with CLIS patients using near-infrared spectroscopy (NIRS) [3,4], leading to many controversies [5] and finally to the retraction of [6] by the editors (due to doubts on the selection within the used data sets and not the applied algorithms; the authors are currently contesting this decision). The main problem is to objectively identify the existence of a consciousness level in CLIS patients, who cannot themselves make this known to the outside world through self-expression. In contrast to other studies that attempt to seek methods of classifying between different clinically defined levels of consciousness, such as LIS, minimally conscious state (MCS) and vegetative state (VS) [7,8], our aim is to identify patterns that are most likely indicative of a minimum level of consciousness that will allow successful communication with an ALS patient in CLIS.

This study proposes to apply multiscale sample entropy (MSE), multiscale permutation entropy (MPE) and multiscale Poincaré (MSP) plots to analyze continuously recorded electrocorticography (ECoG) signals from a CLIS patient in an attempt to uncover whether the patient is experiencing periods of consciousness. The results of the three methods consistently indicated that the CLIS patient was awake during the experiment, in which the patient responded correctly and answered binary yes or no questions with a brain–computer interface system based on ECoG recordings. In this way, we were able to obtain the ground truth from this CLIS patient.

## 2. Materials and Methods

Until today, there are only a few papers discussing the issue of consciousness in ALS patients crossing over into CLIS, since there is no ground truth to validate the results. Our choice of methods was guided by the existing literature on the detection of consciousness in other related physiological and clinical states [9,10], such as anesthesia, minimally conscious state and vegetative state [7,8]. Anesthesia in particular provides a useful means of identifying objective patterns that reflect periods of awareness and unconsciousness, as outlined in the literature. One of the methods that has been used in anesthesia for discriminating between awareness and unconsciousness is sample entropy, focusing mainly on calculating the level of brain wave disorder. It is, thus, reasonable to expect that similar patterns reflecting awareness and unconsciousness will also be observed in a CLIS patient. Based on these expected patterns, we were also able to verify our findings using sample entropy [11,12], permutation entropy [13,14] and the Poincaré plot [15,16], which were found to have relatively high values during the experiment, indicating a status comparable to the awake period of patients in surgery.

The general data processing flow to apply the sample entropy, permutation entropy and Poincaré plot methods is shown in Figure 1. The original ECoG signals were down-sampled to 125 Hz to reduce the processing time, and a sixth-order Butterworth bandpass filter from 1 to 45 Hz was used to remove the power line noise of 50Hz and obtain the filtered signals. Then, sample entropy, permutation entropy and Poincaré plot were applied on the filtered data. Finally, multiscale entropy [17,18] was utilized to reduce the white noise and estimate the consciousness levels. All analyses were conducted using MATLAB R2018b (The Mathworks, Inc.).

### 2.1. Dataset

The dataset was recorded by electrocorticography (ECoG) from a 40-year-old male in a completely locked-in state (CLIS) over 24 h at a sampling rate of 500 Hz with a 64-channel brain amplifier (Brainproducts GmbH, Munich, Germany). He was diagnosed with ALS in 1997 and entered CLIS in 2008. Based on studies by Bierbaumer and colleagues [2,19,20,21], we know that this CLIS patient learned to use this brain–computer interface in an attempt to communicate with the external environment through muscle twitching. The patient had a strong will to live before entering CLIS, but after falling into CLIS, there was only one successful communication out of several attempts. Since the circadian rhythm in CLIS patients is unpredictable, it can lead to a higher frequency of dozing during the day than in healthy individuals [19]. We hypothesized that this lack of communication was a result of the experimenter arriving at the wrong time (i.e., when the patient was not in a consciousness state) and, thus, the patient was unable or unwilling to communicate. Our goal was to identify, if possible, when the patient was in a conscious state and hence most likely to be able to communicate successfully.

Figure 2 (right) shows the position of the ECoG grid electrodes on the left side of the patient’s frontal and parietal lobes. The functional channel locations of the ECoG electrode array are shown in yellow (Figure 2, left panel). The ground channel was S032, and the reference channel was G102 (orange channels, Figure 2, left panel). The implantable electrode was made of material that was difficult to bend; thus, the curvature of the brain could not be fully matched. This probably led to a recording hole in the middle of the electrode array (white channels in Figure 2, left panel) due to channel losses or high impedances. The reference and ground channels were not part of the recording hole.

During the day of recording, an auditory experiment as described in [21] was performed from 14:50 to 17:00. The time window is indicated in grey in Figures 7, 8, 10 and 12. This was the period in which the experimenter was present, and includes both experimental setup and communication with the patient. During the experiment, the patient was expected to answer some paired yes or no open personal questions, such as “You feel good today?” or “You feel bad today?” Personal questions with known answers such as “Are you German?” or “Are you Dutch?” were also asked in order to control the correctness of detection. The patient in question was previously trained to use the brain–computer interface (BCI) while he was in LIS status and was thus aware of how to use the BCI correctly. The patient not only answered the paired questions correctly, but also through questions like “Are you positive regarding the future?”and “Are you negative for the future?” These kinds of questions showed a positive attitude to life during the time of the experiment. We provide as Appendix A the experimental protocol that shows a list of questions and answers during the experiment, in which a + sign means yes and a − sign means no. In order to present the original situation of the experiment, we did not correct the accuracy of the English grammar and the representation of the answers, but excluded the patient’s personal data. We summarized the proportion of correct answers through semantic and contextual judgments, and even if the experimenter asked a paired yes or no question that was not the direction he wanted to go, he expressed his will strongly, and he let the experimenter know the third idea he was trying to express after answering the two opposite meanings. The patient answered 18 yes or no questions during the experiment. One answer was incorrect, and one was unclear, so the CLIS patient had an 88% accuracy rate in the experiment. In the experimental protocol, we indicated the questions with correct answers in green, those with incorrect answers in red and those with unclear answers in black.

### 2.2. Sample Entropy

Sample entropy (SampEn) is based on approximate entropy (AppEn), and it was developed by Richman and Moorman [9] to eliminate some disadvantages of AppEn [22,23]. Compared with AppEn, SampEn is independent from the data length and does not have self-matching. Sample entropy is widely applied in neuroscience and has previously been used to determine the level of consciousness, such as during surgical operations [24] and in some real-time applications [12,25]. To calculate the sample entropy, a time series *X* = [*x*(1), *x*(2),…,*x*(N)] is constructed in which *N* is the data length, which is divided into several subsequences $u_{m}\left(i\right)$, and *m* is the dimension:(1)$$u_{m}\left(i\right)=\left[x\left(i\right),x\left(i+1\right),\ldots ,x\left(i+m-1\right)\right],\;i=1\ldots N-m+1$$

This must meet the following condition:(2)$$d\left[u_{m}\left(i\right),u_{m}\left(j\right)\right]=\max\left\{\left|x\left(i+k\right)-x\left(j+k\right)\right|\right\}<r\times SD$$
where *SD* is the standard deviation of the time series *X* and *r* is the tolerance coefficient. The value of *r* was set to 0.2, and the tolerance would be 0.2 *× SD* (i.e., *x*(*j*) was considered to be consistent with *x*(*i*) if *x*(*j*) met *x*(*i*) in this tolerance). $B^{m}\left(r\right)$ is the summation of the number that *x*(*j*) matches the condition of *x*(*i*):(3)$$B^{m}\left(r\right)={\left(N-m\right)}^{-1}\overset{N-m}{\underset{i=1}{\sum }}B_{i}^{m}\left(r\right)$$

Similarly, we set *m* = *m* + 1 and repeated Equations (1)–(3). $A^{m}\left(r\right)$ is defined as follows:(4)$$A^{m}\left(r\right)={\left(N-m-1\right)}^{-1}\overset{N-m}{\underset{i=1}{\sum }}A_{i}^{m}\left(r\right)$$

SampEn is then defined as
(5)$$SampEn\left(N,m,r\right)=-\log\frac{A^{m}\left(r\right)}{B^{m}\left(r\right)}$$

The more complex the series, the higher the value of sample entropy. On the contrary, the more self-similarity in the series, the lower the value of sample entropy. The number of *N* was suggested as the average sum of a minimum $10^{m}$ and a maximum $30^{m}$ by Richman and Moorman [9] and Pincus and Goldberger [26]. Thus, the parameter *m* = 3 was selected to meet the relationship between *N* and *m*.

### 2.3. Permutation Entropy

Sample entropy is computationally expensive [15], and this is important for the future development of a medical device. Therefore, we also investigated the less computationally expensive method of Permutation entropy and compared the results with those from sample entropy.

Mathematically, permutation entropy (PE) does not consider the exact values of a time series, but instead considers the ordering of the time series, thus reducing the complexity of its computation [27,28]. The time series is divided into several subsequences with a length of *m* epochs, and each sample is converted into a series of rankings (patterns). An example of PE computation is shown in Figure 3. In Figure 3a,b the time series *X* was divided into the values (12,9,7), (9,7,6), …, (10,5,13), and converted to the patterns (3,2,1), (1,2,3), …, (2,1,3). Figure 3b indicates the principle templates of the data sequences and then sums up the number of permutation pattern appearances of each observed sequence. It is different to SampEn since SampEn uses tolerance to ensure whether two time series have similar patterns, whereas PE has a fixed number of *m**!* ordinal patterns. The complexity of PE is defined by *m*, with the possible permutation patterns estimated as *m**!*. Figure 3c shows the relative frequency of occurrence of all possible ordinal patterns for this example.

Riedl et al. [13] suggested that a value of *m* between 3 and 7 was more appropriate for electroencephalogram (EEG) applications. In order to save computation time, we used a relatively low value of *m* = 3, although it may have reduced the sensitivity of the result.

### 2.4. Poincaré Plot

The Poincaré plot is a non-linear geometrical representation of successive measurements providing a visual representation of time series variability. A common application of Poincaré plots is the detection of the short-term and long-term variability of a heart rate [29,30], but in recent years, Poincaré plots have also been used in neuroscience applications, such as to detect the depth of anesthesia [15,16,31]. For our data set, we used pairs of scatterplots of each ECoG voltage *x*(*n*) versus the next ECoG voltage after a time delay Δ*x*(*n* + Δ) for the generation of each Poincaré plot. To quantify the distribution of the ECoG signals in the Poincaré plot, the standard deviation (SD) perpendicular to the diagonal line (SD_1_) and the SD along the diagonal line (SD_2_) were measured, as shown in Figure 4. Its mathematical expression is as follows: SDX is the standard deviation of time series, and SDSD is the standard deviation of the succeeding difference of time series.
(6)$$SD_{1}^{2}=\frac{1}{2}SDSD^{2}=\gamma_{X}\left(0\right)-\gamma_{X}\left(1\right)$$
(7)$$SD_{2}^{2}=2SDX^{2}-\frac{1}{2}SDSD^{2}=\gamma_{X}\left(0\right)+\gamma_{X}\left(1\right)-2{\bar{X}}^{2}$$
where $\gamma_{X}\left(0\right)$ and $\gamma_{X}\left(1\right)$ are the autocorrelation function for the lag-0 and lag-1 ECoG time series and $\bar{X}$ shows the mean of the ECoG time series.

In this study, the Poincaré plots from 30 s epochs of the ECoG signal (the sampling rate was 125 Hz) were plotted. The time delay was thus set to 1/125 s, which must be around one-fifth to one-fourth of the dominant cycle period or a multiple of the signal sampling interval. The choice of the optimum time delay could exactly reconstruct the underlying characteristics of the system [15].

### 2.5. Multiscale Approach

Costa et al. [17,18] reported that multiscale approach analysis, according to SampEn estimations of the heart rate, was used. Before using the multiscale approach, white noise was assigned to a higher entropy value than pink noise. However, when the scale size increased, the entropy value of the coarse-grained white noise decreased. Despite this, the change of the scale size had no influence on the entropy value of the coarse-grained pink noise, which was almost constant. If the scale size was more than 4, the entropy value of the white noise was lower than the corresponding value of pink noise. Therefore, in our study, we chose the scale size equal to 4. Figure 5 shows the schematic diagram in which the time series b_1_, b_2_, …, b_i_ was created by separating the original time series a_1_, a_2_, …, a_i+3_ into non-overlapping windows of a scale of 4 and then averaging the time series in each window.

## 3. Results

In this paper, three commonly used methods in neuroscience applications—multiscale sample entropy (MSE), multiscale permutation entropy (MPE) and multiscale Poincaré (MSP) plots—were applied in order to identify the periods of consciousness in this CLIS patient.

### 3.1. Multiscale Approach

Figure 6 and Figure 7 show the results before and after the multiscale approach for 24 h, where the result before the multiscale approach is very noisy. In order to obtain the main trend and reduce the effect of white noise, we utilized the multiscale approach after sample entropy, permutation entropy and SD_1_ of the Poincaré plots to extract a cleaner result from all approaches.

### 3.2. Multiscale Sample Entropy

As proposed in [21,32], the results of the sample entropy were utilized to analyze consciousness. A higher value of MSE indicated increased complexity within the ECoG signals, which was more indicative of periods of consciousness. The average MSE from all 59 usable channels over the 24 h recording period is shown in Figure 7. As we know from the experiment performed the day of recording, the patient answered the questions correctly during the auditory experiment, which took place from 14:50 to 17:00. The average value of MSE during the experiment was 1.5242. Using this information as the ground truth, and thus setting the consciousness threshold value at 1.52, the periods between 15:18–15:56, 16:04–16:22 and 16:52–17:02 could be labelled as periods of consciousness. Under this premise, we could label the periods between 08:28–08:36, 11:12–11:42, 11:50–12:02, 23:40–01:34, 01:46–02:32 and 03:06–03:16 as periods of consciousness as well.

### 3.3. Multiscale Permutation Entropy

In order to reduce the computation time, we also applied multiscale permutation entropy (MPE) in this study to analyze the consciousness state in the time domain. Smaller MPE values imply increased pattern similarity in the time series and, hence, less complex brain activity. Figure 8 shows the average MPE from all 59 usable channels over 24 h. The average value of MSE during the experiment was 2.388. Using this as a consciousness threshold, we could identify the periods between 15:46–16:10 and 16:26–16:50, but also between 09:06–09:42, 09:58–10:10, 10:46–11:02 and 12:26–12:58, as periods of consciousness.

### 3.4. Multiscale Poincaré (MSP) Plots

Golińska [30] reported that the Poincaré plot showed a fluffy [16] pattern during light anesthesia and an elongated pattern during deep anesthesia. We used this interpretation in our application, i.e., if the CLIS patient was conscious, a fluffy pattern was displayed in the results of the Poincaré plot, as shown in Figure 9 (left). In contrast, if the CLIS patient was unconscious, the results of the Poincaré plot displayed an ellipse pattern, as shown in Figure 9 (right). To distinguish the difference, SD_1_ represents the instantaneous variability, and SD_2_ shows the ECoG voltage variability of the 30 s recoding time window. We show the 24 h results of SD_1_ and SD_2_ in Figure 10 and Figure 11.

To reduce white noise in the result of the Poincaré plot, we utilized the multiscale approach, which is an approach also described by Henriques for heart rate variability [29]. The SD_1_ and SD_2_ of the MSP plots for 24 h are shown in Figure 10 and Figure 11. The result of SD_1_ was the average of all 59 usable channels, and SD_2_ was the average of 52 usable channels. The trend of SD_1_ and SD_2_ was similar to MSE. The distributions of MSP plots during the experiment had the high-peaked and heavy-tailed characteristics, which are different from the flat-topped characteristic of MSE and MPE. The kurtosis of MSE distribution was leptokurtic, and the MSE and MPE distributions were platykurtic. Considering the preparation during the experiment, we raised the threshold to the average of the upper half of the sorted data during the experiment. With a threshold value of SD_1_ at 4.36, the ground truth periods at 16:04–16:22 and 16:56–17:04 exceeded the threshold. Using this consciousness level threshold, the periods between 10:22–10:44, 11:24–12:14, 01:08–02:46, 03:10–03:44 and 04:46–05:04 could be labelled as periods of consciousness. Both SD_1_ and SD_2_ showed similar trends, but the result of SD_1_ was clearer, so we will use it to compare it with the other methods.

## 4. Discussion

The results of the three different approaches proposed in this paper suggest that this CLIS patient was conscious between approximately 16:04–16:10. This time window corresponds to the period during the experiment in which the CLIS patient correctly answers the pairwise yes or no questions via a BCI [2,21].

The results of the MSE and MSP plots indicate that this patient was probably awake during the night, supporting the result of [19], which reported that sleep fragmentation of slow wave sleep (SWS) increases during the process of ALS patients slipping into CLIS.

The trend of the results was similar for the MSE and MSP plots, but different from MPE. Permutation entropy represents the time series in *m!* possible ordinal patterns, attenuating the information from voltage variation and thus reducing the sensitivity of the result. It is possible that the CLIS patient dreams during the night. The voltage variations are large, but the ordinal patterns are similar because there are fewer environmental stimuli to induce brain activity when the patient sleeps at night. It should also be noted that during dreaming, one can state there is consciousness.

Figure 12 shows the result of the majority decision of the three methods over a period of 24 h. From 16:04 to 16:10, the MSE, MPE, and MSP plots show that this CLIS patient was in a conscious state. Even though all methods indicated more or less precisely the reported time of consciousness of the patient by the experimenter that day, the exact times differed from one method to another. In addition, other time periods of detected consciousness were not consistent over all methods. Thus, the level of sensitivity of the applied methods can be defined as being different and thus more or less responsive to the change of consciousness. This indicates that a single method may not be an appropriate approach to identify exactly the levels of consciousness in CLIS patients. Currently the definition of the threshold is based only on the value of each method in the time window of the experiment that the patient communicated with the investigators. In the future, when there are more cases of this type of disease with a proofed moment of consciousness, we hopefully can define the threshold for consciousness and unconsciousness in CLIS patients in a more general way.

Because the experimental period was 14:50–17:00 and there was an 88% accuracy response rate, we used the 15:00–17:00 experimental period to compare the distribution of values with all other 2 h non-experimental slots. We used the Z-test, and since the value ranges of the individual methods were different, we normalized the value of the individual methods to (0,1). The result for MSE is shown in Figure 13a. The Z-test revealed that the multiscale sample entropy (MSE) value was significantly different in all non-experimental periods compared with the experimental period, except for the nighttime period, and the higher MSE values could probably be attributed to the rapid eye movement (REM) during dream sleep, which indicates consciousness.

Figure 13b shows the statistical results for multiscale permutation entropy (MPE), illustrating that there is no significant difference in the 09:00–11:00 time period. Since the multiscale Poincaré (MSP) plots also showed the same results, and the majority decision of all three methods (Figure 12) had higher values, the patients may have been conscious in the morning.

Although the statistical results of the multiscale Poincaré (MSP) plots in Figure 13c showed higher values at night, the Z-test showed significant differences at these time periods compared with the experimental period. The equations of the Poincaré plots were not tolerant, compared with the sample entropy and permutation entropy methods, so the rapid eye movement (REM) during the sleep phase may have led to higher values of the multiscale Poincaré (MSP) plots, as shown by the values. Please note that during the probable REM phase, the values were significantly higher than during the experimental phase, which is in accordance to the relation between consciousness and dreaming. Therefore, combining the results from a number of methods, as we have done in this work, leads to a higher likelihood of correctness.

## 5. Conclusions

Three approaches to detecting the state of consciousness in a complete locked-in patient for 24 h were presented in this paper. Different approaches exhibited different healthy interpretations of consciousness. Although each method suggested different time periods for consciousness, all of them were able to identify the time period around the time window of the experiment where consciousness was confirmed by experimenters receiving correct feedback from the CLIS patient during that period. Therefore, we conclude that for the detection of consciousness in CLIS patients, in order to find appropriate communication times, the combination of the presented approaches in a brain–computer interface system increases the probability of correctly detecting the state of consciousness in CLIS patients, as proposed in [21]. In addition, such a hybrid method approach can even be ameliorated by introducing further methods. Once this can also be shown for other CLIS patients showing verified consciousness periods, it can be considered as a step toward reliable consciousness detection and, thus, a preferred communication option for CLIS patients through a brain–computer interface (BCI).

## Figures and Tables

**Figure 1 entropy-22-01411-f001:**
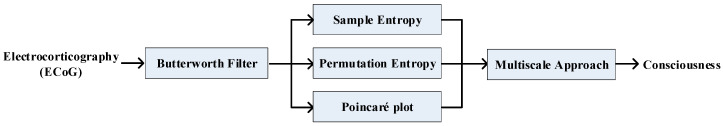
The data processing flow chart.

**Figure 2 entropy-22-01411-f002:**
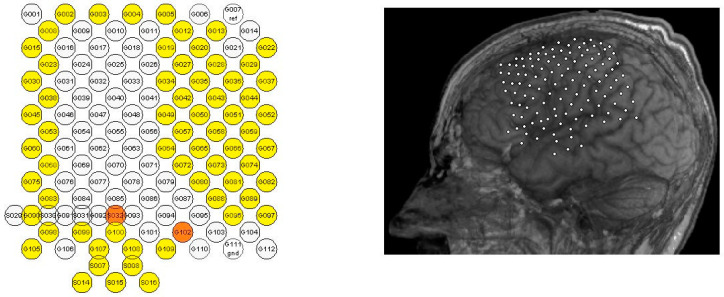
Channel locations of the electrocorticography (ECoG) electrode array. (**Left**): channel names, with the functional channels shown in yellow and the ground channel (S032) and reference channel (G102) in orange. (**Right**): the position of the implanted ECoG grid electrodes.

**Figure 3 entropy-22-01411-f003:**
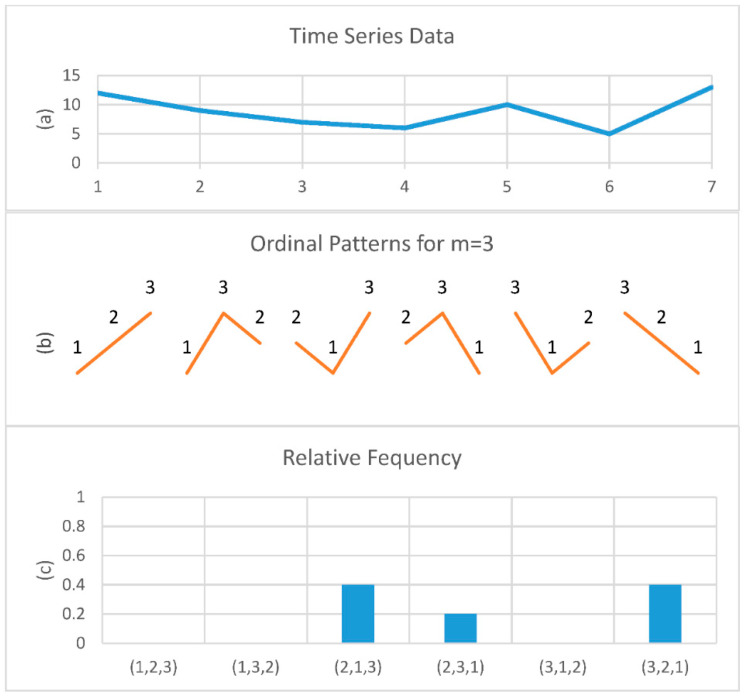
Example of permutation entropy estimation. (**a**) Time series X = (12, 9, 7, 6, 10, 5, 13). (**b**) The six possible permutation patterns for *m* = 3. (**c**) The relative frequency of all possible ordinal patterns for *n* = 3 for this time series, P (2,1,3) = 0.4, P (2,3,1) = 0.2 and P (3,2,1) = 0.4.

**Figure 4 entropy-22-01411-f004:**
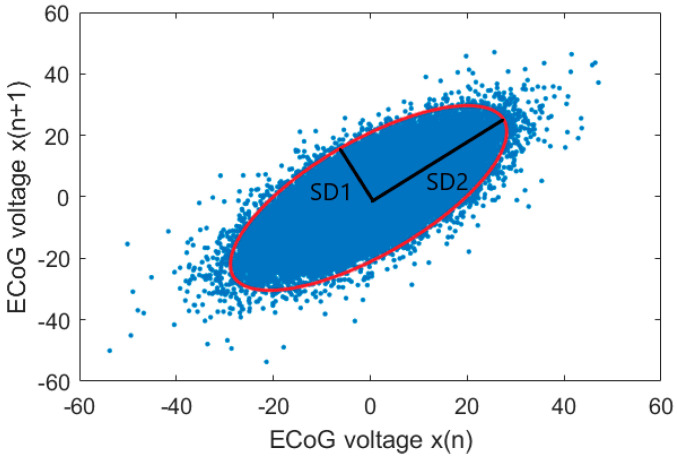
The Poincaré plot (time delay = 1), standard deviation perpendicular to the diagonal line (SD_1_) and standard deviation along the diagonal line (SD_2_) describing the fitted ellipse of the ECoG voltage dispersion along the minor and major axis.

**Figure 5 entropy-22-01411-f005:**
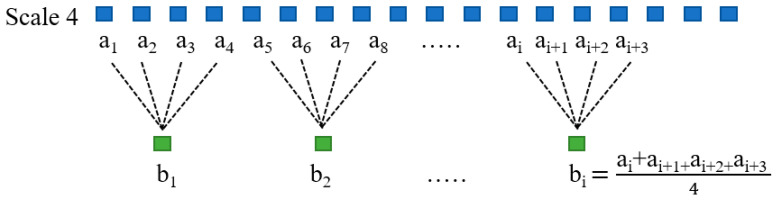
Coarse-graining procedure for a scale of 4.

**Figure 6 entropy-22-01411-f006:**
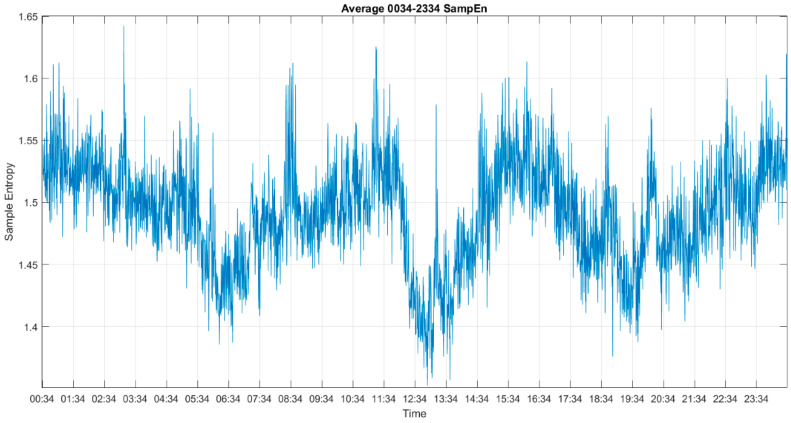
The results of sample entropy for 24 h.

**Figure 7 entropy-22-01411-f007:**
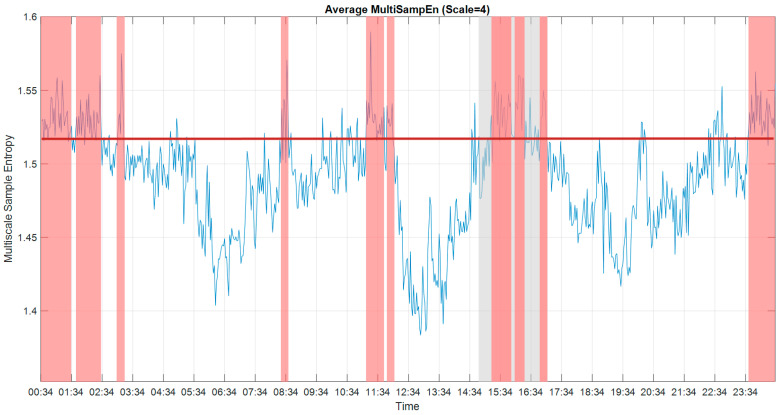
The results of multiscale sample entropy (MSE) (scale = 4) for 24 h. The period of the experiment is indicated by the grey block. The threshold value is indicated by a red horizontal line. The periods of presumed consciousness are indicated by the high values of MSE, shown in red blocks. To avoid the plot being too crowded, we marked the predicted conscious periods, which are predicted conscious periods of more than 6 min for every 10 min overlapping time window.

**Figure 8 entropy-22-01411-f008:**
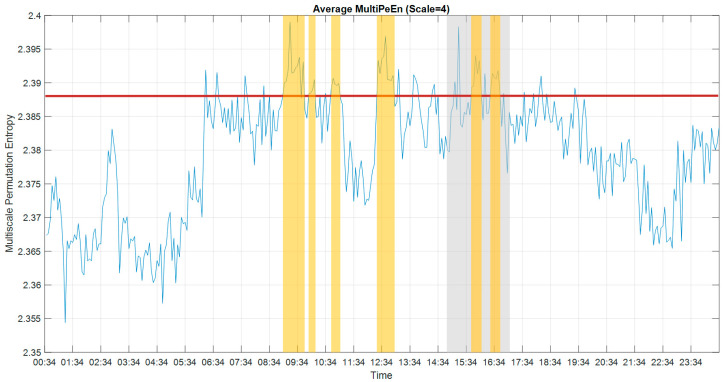
The results of multiscale permutation entropy (MPE) (scale = 4) for 24 h. The period of the experiment is indicated by the grey block. The threshold value is shown by a red horizontal line. The periods of consciousness as presumed by the high values of MPE are indicated in the yellow blocks. To avoid the plot being too crowded, we marked the predicted conscious periods, which are predicted conscious periods of more than 8 min in every 12 min overlapping time window.

**Figure 9 entropy-22-01411-f009:**
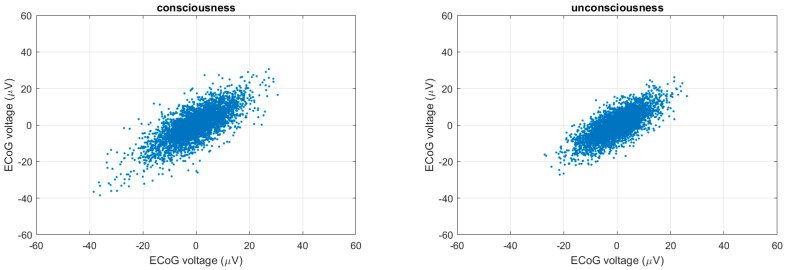
The Poincaré plot. (**Left**): 30 s during the experiment. (**Right**): the other 30 s during the day.

**Figure 10 entropy-22-01411-f010:**
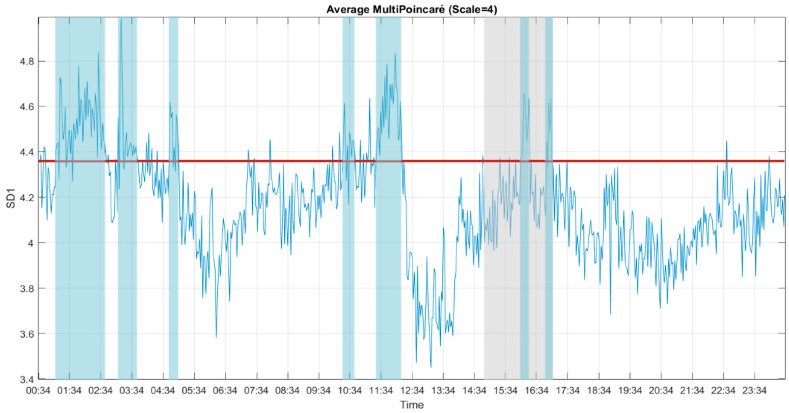
The SD_1_ of the multiscale Poincaré (MSP) plots (scale = 4) for 24 h. The period of the experiment is indicated by the grey block. The threshold value is indicated by a red horizontal line. The presumed periods of consciousness, as indicated by high values of SD_1_, are shown in blue blocks. To avoid the plot being too crowded, we marked the predicted conscious periods which were more than 6 min for every 10 min overlapping time window.

**Figure 11 entropy-22-01411-f011:**
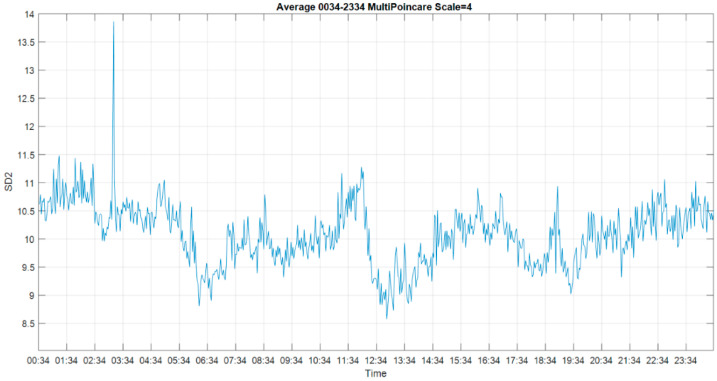
The SD_2_ of the multiscale Poincaré (MSP) plots (scale = 4) for 24 h.

**Figure 12 entropy-22-01411-f012:**
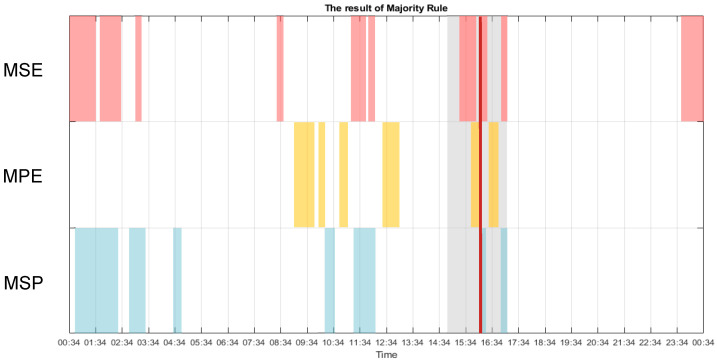
The result of the majority decision of the three methods is indicated by the dark red block. The period of the experiment is shown in the grey block. The state of consciousness identified by the different methods is enclosed by magenta (MSE), yellow (multiscale permutation entropy (MPE)), and blue (multiscale Poincaré (MSP)) blocks.

**Figure 13 entropy-22-01411-f013:**
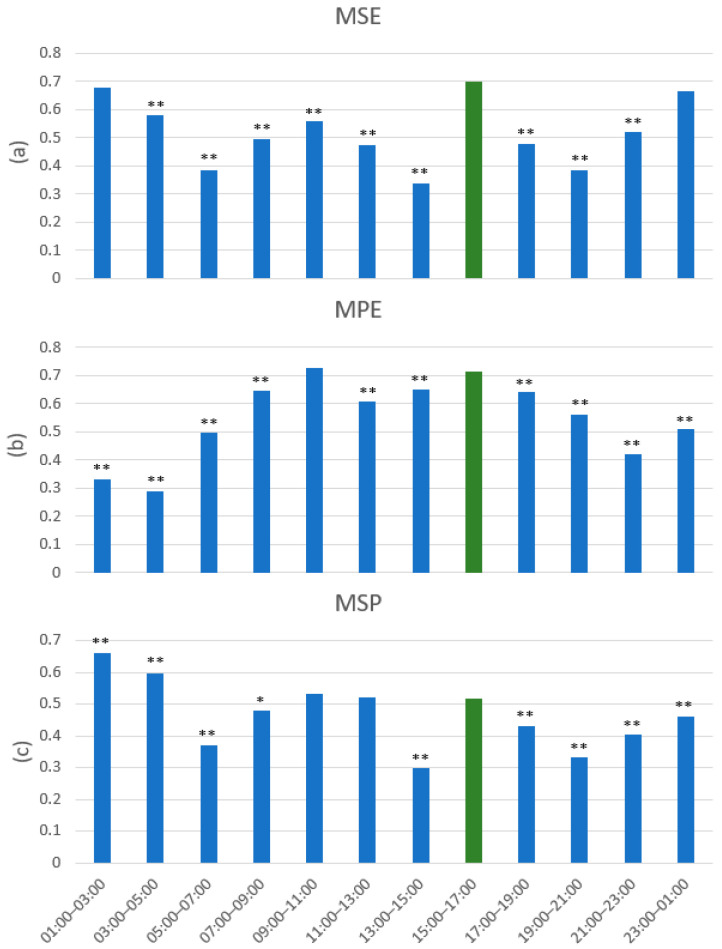
The statistical results (Z-test) for (**a**) MSE, (**b**) MPE, and (**c**) MSP over 24 h. The period of the experiment is shown in green. Significant differences are identified by * *p* < 0.05 and ** *p* < 0.01 in respect to the values from the proven communication time.

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
