# Peer review of "Consciousness Detection in a Complete Locked-in Syndrome Patient through Multiscale Approach Analysis"

_entropy, 2020, doi:10.3390/e22121411_

Round 1

Reviewer 1 Report

The authors report the results of an application of various metrics to ECoG data of a lock-in-sindrome patient. The topic of the investigation is extremely interesting. However, I find the current manuscript unsatisfactory for reasons that I proceed to explain.

First, the goal of the investigation is not entirely clear. The authors state that their aim is to develop tools to quantify conscious level. However, it is not entirely clear what exactly is meant by that, and most importantly, how the findings could be validated.

The study is in general very much method-driven, but the logic behind the applications of those methods is not explicitly developed. In particular, the authors should explain how they choice the employed methods, and what do they expect from them. The methods section should motivate the choice of methods, and delineate what are the expected results considering the available literature.

Additionally, the reported results are just anecdotal. First, it is not clear why an increase of these methods should correlated with states of higher conscious level. Additionally, there should be a more clear method to validate the results. Finally, the lack of any statistical analyses make the results difficult to be evaluated.

Other suggestions:

- It is not clear how the threshold levels are selected. There should be a principled way of this, as they play an important role in the analysis.

- It is not clear how the multiscale approach is employed. How much coarse-graining was used? How was that level selected?

- In Figure 1, the multiscale approach is not a posterior stage, but is modulating the sample entropy, permutation entropy, and poincare plot boxes, no? Also, why transfer entropy is not included in Figure 1?

- Was the multi-scale approach used in combination with transfer entropy? If not, why?

- The analyses of the agreement of the various methods should be included in the results, and not in the discussion. Probably, it could be interesting to have a last box in Fig.1 which compares the various metrics, and possibly combine them to make an improved decision.

- The paper should mention some important milestones in the study of lock-in patients such as [1] and [2].

[1] Owen, A. M., Coleman, M. R., Boly, M., Davis, M. H., Laureys, S., & Pickard, J. D. (2006). Detecting awareness in the vegetative state. science, 313(5792), 1402-1402.

[2] Casali, A. G., Gosseries, O., Rosanova, M., Boly, M., Sarasso, S., Casali, K. R., ... & Massimini, M. (2013). A theoretically based index of consciousness independent of sensory processing and behavior. Science translational medicine, 5(198), 198ra105-198ra105.

Reviewer 2 Report

This paper presents an interesting approach to identify periods of high consciousness in CLIS patients, who in general have fluctuating awareness levels that cannot be assessed at the behavioral level. The paper could have important clinical implications, however, some further clarifications are required.

Major issues:

In the abstract and introduction, there seems to be some misunderstandings regarding the definition of complete locked-in patients and related to that, the aim of the current study. The aims to “detect the underlying consciousness level” and “create reliable communication with CLIS patients” are mentioned like interchangeable, while in reality they are not. First, it needs to be established that the patient is truly CLIS rather than suffering from a disorder of consciousness, and only then covert communication can be established. It seems the authors consider that the only reliable test of consciousness is to establish communication. This is a rather particular view that should be supported by references. In patients with disorders of consciousness there is a wide range of behaviors that suggest the presence of consciousness, of which reliable communication is a high-level one (as described in the JFK Coma Recovery Scale-Revised by Giacino, Kalmar and Whyte, 2004). However, the establishment of functional communication could also signify that a patient is emergent from the minimally conscious state, rather than suffering from a LIS. The diagnosis is LIS is usually also supported by either the etiology (like ALS) or the type of brain lesion (classically at the pons, assessed through neuroimaging). This should be clarified throughout the introduction. Furthermore “The main problem is to objectively identify the existence of a consciousness level in CLIS patients, who cannot themselves make this known to the outside world through self-expression” (line 48-49) is not the main aim of the paper and confusing the reader. From further reading it becomes clear that the aim of the paper is to identify moment of presumed high awareness within the fluctuating levels of consciousness that patients with severe brain injury usually present. These moments of high awareness are identified in the current approach as a moment in which communication (part of another paper) is established. The introduction would benefit from rewriting to clarify these issues.

More information regarding the patient could be provided. What was the etiology of the patient, how long has he been in this state? How was the diagnosis established (Line 75)? Especially the last point seems crucial, as the authors suggest in their manuscript that the diagnosis can only be made by establishment of communication.

From a clinical perspective it would be great if the authors could provide the list of questions and answers (excluding the personal ones that would de-anonymize the patient) that were obtained, either in the main manuscript or supplementary material. This would (1) help to establish how many questions were used to assess the consistency in the patient’s replies to the paired yes/no questions. (2) Shed light on the patient’s attitude regarding quality of life, which is a timely and hardly investigated topic with great medico-ethical implications. If published elsewhere the authors could refer to that.

Where these measures for consciousness applied online or offline? And if they were applied offline, would it be feasible to obtain them in real-time, so they can be employed in clinical practice?

The identification of “high consciousness” periods is done based on task-related ECoG. How are the authors certain that this translates also to resting state ECoG, as applied to the rest of the 24-h recording?

Minor issues:

The last paragraph of the introduction seems superfluous and could be removed, as the manuscript follows a structure that is the standard for scientific research papers.

Could the authors specify whether (part of) this dataset has been published before, other than reference 2?

Line 87: The verb “do” seems missing in the example questions: "You feel good today?"/"You feel bad today?"

Could the authors specify whether the reference and ground were part of the “recording hole”?

In figure 5 there are red marks under “Scale 4”.

The typo conscous occurs several times.

In the acknowledgement the name hc. mult. should be written with capitals.

Round 2

Reviewer 1 Report

I congratulate the authors for a good job in addressing my comments. At this stage I am only slightly confused with parts of Section 4, which I hope can be clarified before publication. I hope these new comments may help to improve the final version of the paper.

First, while it seems clear that the patient had some level of awareness from 14:50-17:00, what can be said about the time outside this period?

Related to that, how should the periods of relative agreement of the metrics outside that period (shaded in Fig 12) be understood? Is that a false positive of the metrics?

Also, could the authors clarify how the three metrics should be combined for doing an assessment of the patient? Should one take the min of three, or something like that?

Also, in the response the authors say that statistics cannot be calculated because this is a single subject. But could be possible to do within-subject stats? Eg, instead of the comparison presented in Table 1 with a single non-experiment period which feels arbitrarly picked, could the authors make a histogram of the values of the quantities in all non-experiment periods and see how is it related to the actual values?

Reviewer 2 Report

I would like to thank the authors for the clarifications and additional (mostly clinical) information provided. The report of the conversation with the patient is really unique and valuable. I have no further comments, other than that I hope this work will severe several LIS patients in the future to identify (potential) periods of high responsiveness. All too often this kind of work does not make it to the clinic.

Author Response

Dear Reviewer,

Thank you again for your time in reviewing our manuscript and for your valuable comments, which have helped us to improve our work. Please see the revised paper for minor revisions. Changes are also indicated in red in the revised paper. Thanks for you wishes, we hope, that we may help the patients indeed and finally come to be in clinical application. You’re right, too often these approaches doesn’t make it through….

Best regards

Shang-Ju Wu, Nicoletta Nicolaou, Martin Bogdan